# Legolas: Deep Leg-Inertial Odometry

**Justin Wasserman**[1,2], **Ananye Agarwal**[2], **Rishabh Jangir**[2],
**Girish Chowdhary**[1], **Deepak Pathak**[2*], **Abhinav Gupta**[2*]
[1]University of Illinois at Urbana-Champaign,
[2]Carnegie Mellon University
*

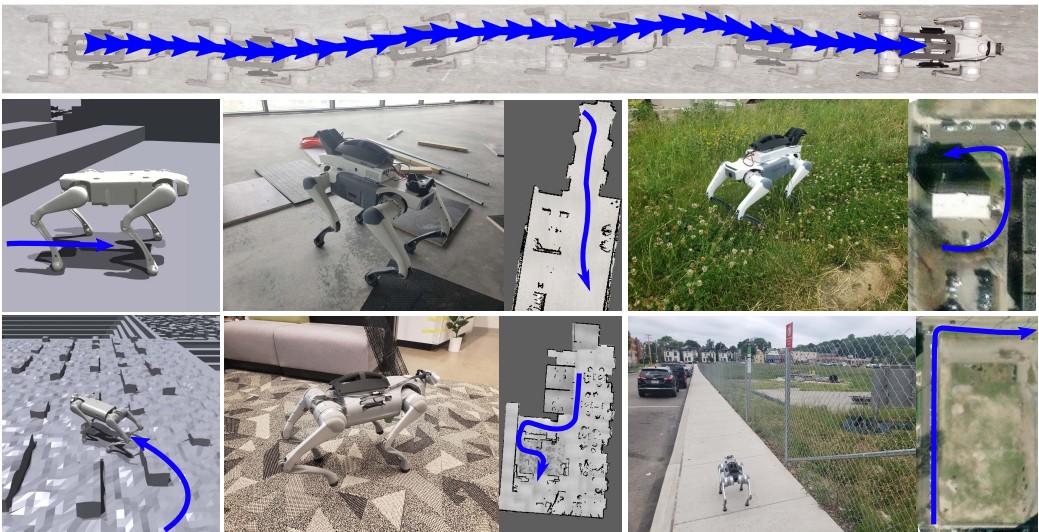

Figure 1: **Robot state estimation by Legolas across diverse environments and robotics platforms.** Legolas operates at a high frequency to predict small incremental motions of a robot by only utilizing leg and IMU sensing. We introduce the first accurate, purely data-driven method for predicting odometry.

**Abstract:** Estimating odometry, where an accumulating position and rotation is tracked, has critical applications in many areas of robotics as a form of state estimation such as in SLAM, navigation, and controls. During deployment of a legged robot, a vision system's tracking can easily get lost. Instead, using only the onboard leg and inertial sensor for odometry is a promising alternative. Previous methods in estimating leg-inertial odometry require analytical modeling or collecting high-quality real-world trajectories to train a model. Analytical modeling is specific to each robot, requires manual fine-tuning, and doesn't always capture real-world phenomena such as slippage. Previous work learning legged odometry still relies on collecting real-world data, this has been shown to not perform well out of distribution. In this work, we show that it is possible to estimate the odometry of a legged robot without any analytical modeling or real-world data collection. In this paper, we present Legolas, the first method that accurately estimates odometry in a purely data-driven fashion for quadruped robots. We deploy our method on two real-world quadruped robots in both indoor and outdoor environments. In the indoor scenes, our proposed method accomplishes a relative pose error that is 73% less than an analytical filtering-based approach and 87.5% less than a real-world behavioral cloning approach. More results are available at: learned-odom.github.io.

**Keywords:** State and Odometry Estimation, Quadruped robots, Sim-to-Real

---

*\*Equal Advising

8th Conference on Robot Learning (CoRL 2024), Munich, Germany.

# 1 Introduction

Many robotics tasks require tracking the state of the robot, one of these states that is commonly tracked is the pose of the robot. Doing so with odometry requires estimating and accumulating small incremental ego-centric displacements of the robot at a high frequency. Within robotics, these estimates are used in many downstream tasks. An example application is within SLAM, where odometry estimates are used as motion estimates to improve mapping [1]. It is also common in controls to use odometry estimates as a source of feedback [2, 3]. Odometry has applications across various fields where fast tracking is required. Some notable examples include VR [4], pedestrian motion estimation [5], and autonomous car navigation [6].

A typical deployment of a robot will often incorporate visual components into its odometry [7, 8, 9, 10]. These components are useful for creating a map to reduce drift and to assist when other sensors fail. However, in many cases, a quadruped will need to act in an environment where it is required to perform quick movements such as going up and down stairs or dodging obstacles. These quick motions cause the visual component to lose tracking and cause the odometry system to fail. During these failure modes we seek to estimate odometry through just the proprioceptive and inertial sensing that are ubiquitously found onboard quadruped platforms. Fig. 1 and the accompanying videos show examples of our robots estimating odometry across many environments.

Previous methods for solving leg-inertial odometry can be split into analytical and learned methods. Analytical methods for estimating odometry on a legged robot often require direct sensing or estimating the foot contact [11, 12, 13, 14]. However, foot contact sensors are not always available on robots, and estimating the foot contacts can be challenging due to modeling the robot's interactions with the environment such as slippage, and deformable surfaces. Due to the analytical models only being correct for a studied robot, immediate transfer to a different robot configuration may not be possible. On the learning-based side, recent methods have shown that it is possible to estimate odometry by training a model from real-world data [15, 16, 17]. However, collecting a diverse and large dataset of trajectories with high-quality labels is challenging. Challenges include making the training environments diverse, limitations of devices that can be used to obtain the target pose, and time to collect the dataset. These systems have also previously relied on analytical models and utilized the prediction in their update step rather than only using the learned estimate.

In this work, we introduce Legolas, or **Leg**ged **o**dometry **l**earned **a**s **s**imulated as a purely data-driven solution for solving leg-inertial odometry. In contrast to analytical filtering methods and recently developed learned odometry prediction models, our proposed work does not require analytical modeling such as sensing modeling, kinematic modeling, or foot contact estimation.

This work builds off the recent success in robot locomotion and navigation [18, 19, 20, 21] in sim-to-real transfer. These works demonstrated that complex planning, mapping, and analytical modeling are not necessary for deploying their methods to real-world platforms. We extend this line of work by showing that predicting the robot's odometry can be successfully learned from simulation as a first step toward sim-to-real transfer of explicit state estimation. Through simulation, months' worth of trajectory data across diverse scenes are collected in a single day. After training, a direct sim-to-real transfer of odometry prediction on the Unitree Go2 and DEEP Robotics Lite3 quadrupeds is demonstrated. After training individual models for each of the quadrupeds, no additional manual tuning is required for deployment. This is in contrast to many current filtering-based and factor graph methods for predicting odometry that suffer from the "Curse of Manual Fine-Tuning" [22]. Furthermore, Legolas can predict odometry at a high frequency at an accuracy much higher than previous filtering and learned baselines while being competitive with a visual odometry method in indoor scenes.

# 2 Related Works

**Filtering-Based State Estimators:** An Extended Kalman Filter (EKF) is a class of filtering-based methods that use non-linear analytical modeling of the robots' kinematics and sensors to estimate the

state of the robot [23]. EKF methods [24, 25] for tracking the state of legged robots are promising as the roll and pitch orientations are fully observable and drift-free on quadrupeds [26]. However, the models to build the EKF often assume that at least one foot is in contact with the ground at all times [12]. Additionally, for quadruped models, it is often assumed that there is no or little slippage on the floor [12, 27]. In contrast to these filtering-based methods, Legolas makes no assumption about analytical modeling, meaning that Legolas can be used to train an odometry model on any robot configuration. Furthermore, Legolas makes no assumptions about the environment as Legolas is trained in simulation, it collects data across many environment variations and configurations. Furthermore, filtering-based methods will often require or model the foot contacts of the robot [28]. As foot sensing is not straightforward to model [29], susceptible to damage through impact [30], and not all pedal robots have this sensor, Legolas does not require the use of this sensor.

**Learning-Based Inertial Odometry:** There is a broad range of literature that is interested in learning odometry through just an IMU [16, 31, 32] as IMUs are cheap and prevalent. These methods have been applied to applications outside of robotics such as consumer phones [4] and drones [33]. Work such as Cioffi *et al.* [33] apply this idea to robotics and have shown that it is possible to use only sensing from an IMU collected from real-world trajectories, in conjunction with a model-based system to predict accurate odometry estimates. However, Buchanan *et al.* [34] has shown that these predictions degrade when the robot's behavior is out of distribution from the training dataset. In contrast to these previous methods, as Legolas is trained in simulation, months worth of trajectories of the robots moving in diverse environments such as over stones, stairs, and inclines are collected a single day without the need for analytical models. This allows us to train a robust odometry prediction model that is less likely to go out of distribution during deployment. Furthermore, in this work, we will show that predictions from just an IMU lead to degraded predictions, and demonstrate superior performance by adding additional observations that are readily available on quadruped robots.

**Robot Sim-To-Real Transfer:** Sim-to-real transfer has been notably of interest recently due to the promising scale of simulators, safety, and the difficulty in collecting real-world data on physical systems [35, 36, 37]. RMA [19] used a two-phase learning procedure to adapt the locomotion policy to unknown environment parameters. Agarwal *et al.* [18] extended the two-phase framework by first training with low-resolution scandots. Then in the second phase, supervised training from the first phase policy is performed with depth images in the observation. Both this method and RMA were demonstrated to transfer to the real world on difficult locomotion tasks such as climbing stairs and moving through slippery environments without fine-tuning. Hoeller *et al.* learns a state representation over a sequence of images and the trajectory of the robot in simulation to perform obstacle avoidance while reaching a goal. This method was shown to even dodge dynamic obstacles after transfer to the real world. Zhuang *et al.* [38] demonstrated successful transfer of highly precise and reactive parkour skills. Adding onto this line of work, Legolas shows a successful sim-to-real transfer of state estimation from simulation across multiple robot embodiments.

## 3   Legolas

The goal is to train a pose difference prediction model $\mathcal{P}$ that predicts small incremental motions of the robot at each time step. This model takes as input a vector of size $\mathbb{R}^{44}$ containing previous actions and command velocities as well as IMU and proprioceptive sensing which are common on quadruped platforms. The IMU data corresponds the gyroscope ($\mathbb{R}^3$) and roll and pitch estimates ($\mathbb{R}^2$). The command velocity is in the x, y, and angular directions ($\mathbb{R}^3$). The proprioceptive sensing corresponds to the joint angle and velocity of each joint of size $\mathbb{R}^{12}$. Finally, the previous actions are given as desired joint angles, this is of size $\mathbb{R}^{12}$. $\mathcal{P}$ outputs a prediction and variance vector of shape $\mathbb{R}^{18}$ on how much the robot has translated ($\mathbb{R}^3$) and rotated as represented by a 6D rotation, $\mathbb{R}^6$ [39]. This representation was chosen as it was found to lead to the lowest error on the validation dataset as discussed in Sec. C. A variance is also predicted for each element of the translation and rotation component $\mathbb{R}^9$. An additional head is introduced to predict confidence ($\mathbb{R}^1$) whether the robot is moving to perform zero-velocity updates [40]. The connection between data collection and training is visualized in Fig. 2a.

A relative pose prediction error is utilized in the proposed loss that compares the predicted and ground truth incremental motions of the robot. A key insight into this loss as visualized in Fig. 2b, is that this error should not be propagated to future predictions. This formulation allows for the creation of independent data points in the collected dataset.

In this section, we will first provide details on dataset collection in Sec. 3.1. After this, we will discuss the loss used to train $\mathcal{P}$ in Sec. 3.2 and finally training details in Sec. 3.3.

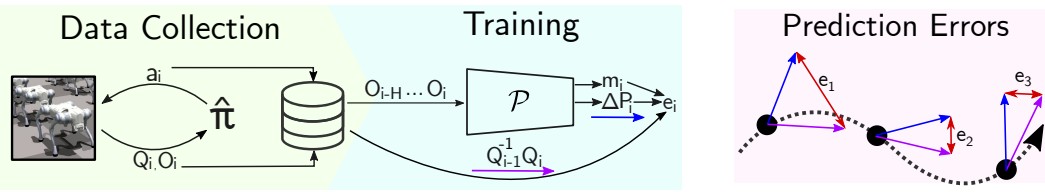

(a) **Overview of data collection and training Legolas.**     (b) **Relative prediction errors.**

Figure 2: **Legolas overview.** (a) 13.7M data points containing a history of the robot's observations and ground-truth pose are collected from simulation. $\mathcal{P}$ takes in a history of observations and outputs the change in the pose of the robot. In conjunction with the ground truth pose difference, an error is calculated. (b) The key idea behind the proposed error is to calculate the magnitude of the difference between the predicted and ground truth pose changes.

## 3.1 Odometry Dataset Collection

In the simulator Issac Gym [35], a trajectory is collected by using a robust locomotion policy $\hat{\pi}$ for $n$ steps in order to ensure diverse dataset collection. The choice of this policy, whether it is a learned or an analytical approach, for example, will be compatible with Legolas. Legolas only assumes that a given policy $\hat{\pi}$, that takes in the current observations and returns an action $a$, is available both in the simulator and the real world. The simulation environment contains many scenes that a robot would encounter such as tripping hazards, uneven surfaces, and stairs. Collected trajectories contain both the current observation $O_i$ and the ground truth pose of the robot. $O_i$ contains the proprioceptive angle and velocity, IMU gyroscope and roll and pitch estimates, the previous commanded joint angles from $\hat{\pi}$, and commanded velocity. While creating this dataset, thousands of robots are simulated to create trajectories in parallel. Data collection for both the Go2 and the Lite3 are the same. Details and visualizations of dataset collection can be found in Sec. B.

## 3.2 Displacement Error and Training Loss

The error is calculated by comparing trajectories represented as a $k$ length sequence of $SE(3)$ matrices where $Q \in \mathbb{R}^{k \times 4 \times 4}$ is the ground truth trajectory and $P \in \mathbb{R}^{k \times 4 \times 4}$ is the predicted trajectory. At a given timestep $i$, the ground truth pose is $Q_i \in SE(3)$ and the predicted pose is $P_i \in SE(3)$. The incremental ground truth and predicted motion from time step $i$ to $i-1$ is $Q_{i-1}^{-1} Q_i$ and $P_{i-1}^{-1} P_i$ respectively. The pose difference, $D \in SE(3)$ between these incremental motions, is defined in (1).

$$D = (Q_{i-1}^{-1} Q_i)^{-1} (P_{i-1}^{-1} P_i) \tag{1}$$

To formulate the proposed loss as independent predictions in the training procedure, the difference in the predicted trajectory ($\Delta P$) is estimated directly from $\mathcal{P}$ as shown in (2). $\Delta P$ is composed of $\Delta P_t \in \mathbb{R}^3$, the incremental motion in the $(x, y, z)$ directions as well as a rotation component $\Delta P_R \in \mathbb{R}^6$. ($Q_{i-1}^{-1} Q_i$) is then converted into a translational and 6D vector, denoted as $\Delta Q$.

$$D = \Delta Q - \Delta P \tag{2}$$

Training with just $D$ as the error, *i.e.* $e = \|D\|_2^2$ leads to an accumulation of drift while the robot is stationary. To counteract this a masking head is introduced into the network. The purpose of the masking head is to learn to manage zero-velocity updates [40]. In practice the masking head performs binary classification on whether the robot is stationary, outputting $m_i = 0$ when stationary and $m_i = 1$ when in movement. Therefore with the masking head, the final error is defined in (3).

$$e = m_i \cdot \|D\|_2^2 + (1 - m_i) \cdot \|(\Delta Q)\|_2^2 \tag{3}$$

The error can be thought of as just $\|(\Delta Q)\|_2^2$ when the robot is predicted to be stationary. This error is minimized when the robot is standing still and penalized when the robot is actually moving. The error is otherwise equivalent to $e = \|D\|_2^2$ when the robot is predicted to be moving.

Training is started with an MSE loss as shown in (4) and then switched to a Gaussian Maximum Likelihood Loss as shown in (5) after training for a sufficient number of samples.

$$L_{MSE} = \frac{1}{N} \sum_{j=1}^{N} \|e_j\|_2^2 \tag{4}$$

$$L_{GML} = \frac{1}{N} \sum_{j=1}^{N} -\log \left( \frac{\exp(-\frac{1}{2} \|e_j\|_{\hat{\Sigma}_j}^2)}{\sqrt{8\pi^3 \det(\hat{\Sigma}_j)}} \right) = \frac{1}{N} \sum_{j=1}^{N} (\frac{1}{2} \log \det(\hat{\Sigma}_j) + \frac{1}{2} \|e_j\|_{\hat{\Sigma}_j}^2) + Cst \tag{5}$$

Where $e_j$ is the error for a given data point in a mini-batch of size $N$, and $\hat{\Sigma}_j$ is the covariance matrix predicted by $\mathcal{P}$ to represent uncertainty. Using the Gaussian Maximum Likelihood Loss improves the motion prediction and facilitates predicting variance. The variance allows for more interpretability of the output of Legolas and facilitates downstream integration with SLAM systems. However, starting training with $L_{MSE}$ was found to improve $\mathcal{P}$ and reduce training instability. Further details regarding these losses can be found in Sec. C.

### 3.3 Training Procedure

After the trajectories are collected as specified in Sec. 3.2, each trajectory is split into $\frac{n}{H+1}$ parts to form an offline dataset to train $\mathcal{P}$. Where $n$ is the length of the trajectory and $H$ is the number of previous observation steps that will be given to the model. Each data point contains the current observations of the robot as well as a history of the last $H-1$ observations before it, $\{O_{i-H}, O_{i-H-1}, ..., O_{i-1}, O_i\}$. These observations provide a history for the model to improve the prediction of the odometry. Finally, each data point also contains the relative pose difference over the ground truth trajectory for the last step, $Q_{i-1}^{-1} Q_i$.

After collecting the ground truth observations, noise is added to each sensor. For the Go2 dataset, 13.7M data points are collected, equivalent to 160 days of data collection. This took approximately 48 hours to collect. During training, for each data point, $\mathcal{P}$ takes in the observations $\{O_{i-H}, ..., O_i\}$ and predicts incremental motion $\Delta P$ and a covariance matrix $\hat{\Sigma}$. During training and deployment, $H = 50$. This is equivalent to taking the last one second of observation history and predicting the incremental motion over the last 0.02 seconds. Consistent with prior work that estimates odometry [15, 17], a 1D ResNet architecture [41] is adopted for $\mathcal{P}$. Further details can be found in Sec. B.

## 4 Experiments

### 4.1 Experimental Setup

Ground truth trajectories are collected on real-world quadruped robots. Ground truth information is collected with motion capture for indoor scenes and RTK-GPS for outdoor scenes. When neither is available due to their limitations, SLAM with a LIDAR [42] is used to obtain the ground truth pose. Legolas is deployed onto the Unitree Go2 quadruped and the DEEP Robotics Lite3 quadruped at 50 Hz to match the simulation training speed. In the following experiments, more focus is given to the Go2 as baselines are more readily available for this platform.

We benchmark using filtering-based, behavioral cloning, and visual baselines and variations:
• EKF [26]: Analytical filter-based approach, stemming from the work of Cerberus2.0 [25]. The kinematic modeling in the EKF is modified to match the Go2 rather than the Go1. Manual fine-tuning on its parameters is performed [43].
• BC: Trained by collecting trajectories in the real world. These trajectories are then processed into a dataset in the same manner as performed in Sec. 3.3 to create a dataset of 90,000 data points. Training also follows the same as Legolas with the same loss in Sec. 3.2.

| # | Method | Indoors | | | Outdoors | |
|---|--------|---------|--------|--------|----------|--------|
| | | RPE@1m $\downarrow$ | $ATE_o \downarrow$ | $ATE_u \downarrow$ | $ATE_o \downarrow$ | $ATE_u \downarrow$ |
| 1 | EKF [26, 25] | 0.38 | 0.116 | 0.062 | 0.23 | 0.07 |
| 2 | BC | 0.81 | 0.175 | 0.099 | 0.20 | 0.12 |
| 3 | Ours w/ Go1 | 0.12 | **0.022** | **0.014** | 0.08 | 0.04 |
| 4 | Ours | **0.11** | 0.026 | 0.015 | **0.04** | **0.03** |
| 5 | VINS-Fusion [7] | 0.09 | 0.022 | 0.011 | - | - |

Table 1: **Legolas vs. prior method on real-world deployment.** In the indoor setting, Legolas achieves performance at a level comparable to a visual baseline and outperforms relevant leg-inertial baselines. However, in the outdoor setting, where the visual baseline fails to deploy, Legolas still surpasses the other leg-inertial baseline.

• VINS-Fusion [7]: Visual-inertial odometry baseline. Additional hardware and manual fine-tuning was required for suitable deployment [44, 43]. See Sec. D for more details. Has access to stereo images, whereas Legolas and the other baselines do not.
• Ours w/ Go1: Legolas, but instead of collecting a training dataset in the simulator with the Go2, a Go1 is simulated and its respective $\hat{\pi}$ is utilized instead.

## 4.2 Metrics

The RPE and ATE metrics [45] are utilized as they are standard in visual odometry and SLAM [46]. The RPE metric is more relevant to odometry as this metric is an average error of the odometry models across predictions. Both metrics are utilized to be consistent with relevant literature [15, 33].

**RPE:** Relative Pose Error is a "local" error that ignores drift between $Q$ and $P$, it is defined in (6).

$$RPE_{\text{rmse}} := \sqrt{\left(\frac{1}{n}\sum_{i=1}^{n}\left\|t((Q_i^{-1}Q_{i+\Delta})^{-1}(P_i^{-1}P_{i+\Delta}))\right\|^2\right)} \tag{6}$$

Where $\Delta$ corresponds to the index of the next pose in the ground truth trajectory that is some distance away from $Q_i$. In the following experiments, a distance of 1 meter is utilized, therefore this metric is reported as RPE@1m.

**ATE:** Absolute Trajectory Error is a "global" error that penalizes drift and mismatches between the ground-truth $Q$ and predicted trajectory $P$ of lengths $n$. In this work, the root mean squared error of the ATE, which is defined in (7) is reported:

$$ATE_{\text{rmse}} := \sqrt{\left(\frac{1}{n}\sum_{i=1}^{n}\left\|t(Q_i S P_i)\right\|^2\right)} \tag{7}$$

Where $t()$ retrieves the translation component of the matrix, $S$ is a transformation that aligns $Q$ and $P$, and $N$ is the number of steps in a given trajectory. $S$ is necessary as the trajectories are not guaranteed to be immediately aligned. In this work ATE is utilized in two manners, the first is when $S$ is a transformation that aligns the first frame of both trajectories. The second is when $S$ is found using Umeyama's method [47] to find the best transformation to align all points across both $Q$ and $P$. These metrics are referred to as $ATE_o$ and $ATE_u$ respectively. In addition to the above, the error is normalized by the path length so the metric is not dominated by long trajectories.

## 4.3 Real-World Quadruped Odometry

Legolas and baselines are deployed in the real world across 11 indoor and 5 outdoor scenes. These scenarios cover a range of locomotion tasks on flat and hilly terrains across diverse trajectory variations. In Tab. 1 the metrics achieved by each method on both indoor and outdoor scenes are reported. In the outdoor scenes, the RPE@1m metric is forgone as GPS does not accurately report the heading.

**Legolas achieves near vision-based performance on indoor scenes (indoors rows 4 and 5).** While Legolas (row 4) does fall behind VINS-Fusion for indoor scenes, on methods with equal

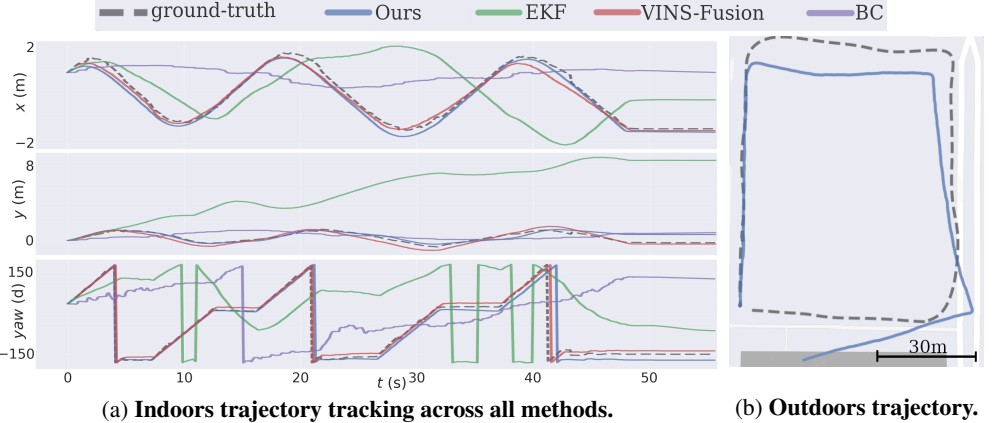

| (a) **Indoors trajectory tracking across all methods.** | (b) **Outdoors trajectory.** |

Figure 3: **Indoor and outdoor trajectories collected on Go2 quadruped.** Across indoor and outdoor scenes, Legolas demonstrates successful deployment. (a) Legolas and VINS-Fusion most closely track the ground-truth. EKF captures a relevant shape but does not track the ground-truth. (b) Legolas produces accurate odometry estimation even in large environments. More full rollouts can be found in Sec. E

sensing, Legolas exceeds the baselines. When compared to the EKF-based approach (indoors row 1), Legolas achieves a 73% lower `RPE@1m`. Furthermore, when compared to the real-world behavioral cloning approach, Legolas achieves an 87% lower `RPE@1m`.

In Fig. 3a, the predicted trajectory across methods is compared. Both VINS-Fusion and Legolas closely track the ground-truth trajectory. EKF does capture a relevant shape but fails to closely track the ground truth over time. We find that this common error mode stems from foot contact estimation being required. However, this estimate comes directly from the foot contact sensor which is prone to noise and degradation [30]. We find that BC can successfully reproduce trajectories it was trained on, but fails to generalize to novel trajectories. This is due to the low amount of data used to train the model when compared to Legolas. This phenomenon has been found in other works as well [34].

**Legolas improves over baselines in the outdoors dataset split (outdoors rows 1, 2, 4)** The outdoors dataset split was notably challenging as there were variations in weather conditions and terrain. Despite this, in Fig. 3b Legolas tracks the robot in a slippery environment, where the ground is covered in rain. Across all baselines in the outdoor dataset split, Legolas improves over the baselines.

VINS-Fusion was tuned for indoor environments and immediate outdoor deployment failed. More fine-tuning didn't improve performance due to brightness changes, using a downward-facing camera, and quick motions. These failure modes are described in-depth along with an example of how Legolas can be integrated with a VIO method to improve robustness can be found in Sec. D.

**Legolas works across embodiment (indoors rows 3 and 4).** While trained with a different physical robot and policy than the one used in deployment, Legolas is capable of accurate odometry prediction in indoor scenes. Notably, when trained with the Go1 and deployed on the Go2, a decrease in $\text{ATE}_o$ from 0.024 to 0.019 and $\text{ATE}_u$ from 0.014 to 0.013 (indoor rows 3 and 4) occurs. However, the more relevant metric for odometry, `RPE@1m`, increases from 0.11 to 0.12.

## 4.4 Simulator Odometry Prediction

In this experiment, a validation dataset is collected in simulation to perform ablations on the observations given to Legolas. This dataset consists of 4 trajectories collected over $90m$ of walking over a rugged ground environment. Results are showcased in Tab. 2.

**Using the full sensor suite and actions history is critical for effective odometry prediction.** Notably, only using the IMU for odometry prediction, yields a `RPE@1m` of 0.32 and an $\text{ATE}_o$ of 0.244 (row 1). Just adding the last actions performed by the robot to the observation space improves

| | Observation Space Modalities | | | Metrics | | |
|---|---|---|---|---|---|---|
| # | Proprioceptic | Last Action | IMU | RPE@1m ↓ | $\text{ATE}_o$ ↓ | $\text{ATE}_u$ ↓ |
| 1 | ✗ | ✗ | ✔ | 0.32 | 0.244 | 0.094 |
| 2 | ✗ | ✔ | ✗ | 0.40 | 0.243 | 0.111 |
| 3 | ✗ | ✔ | ✔ | 0.27 | 0.186 | 0.053 |
| 4 | ✔ | ✗ | ✗ | 0.34 | 0.255 | 0.111 |
| 5 | ✔ | ✗ | ✔ | 0.36 | 0.247 | 0.109 |
| 6 | ✔ | ✔ | ✗ | 0.24 | 0.132 | 0.087 |
| 7 | ✔ | ✔ | ✔ | **0.05** | **0.051** | **0.016** |

Table 2: **Effect of inputs on Legolas.** Thre of the inputs to the Legolas system are Proprioceptic (angle and velocity), the Last Action performed, and IMU. Removing any of the inputs affects the best-performing variation (row 7) of Legolas. When compared to the complete system, the Last Action is most crucial removing this yields an increase in $\text{ATE}_o$ of $484\%$ (row 5), followed by Proprioceptic with a $364\%$ larger error (row 3), and then IMU with an increase of $258\%$ (row 6). Predicting with just the IMU (row 1) as done in previous methods leads to a 478% increase of $\text{ATE}_o$.

this result to a RPE@1m of 0.27 and an $\text{ATE}_o$ of 0.186, a 15.6% and a 23.7% decrease respectively (rows 1 and 3). Finally, adding the proprioceptic sensing to the observation space yields a RPE@1m of 0.05 and an $\text{ATE}_o$ of 0.051, an incredibly large 81.4% and 72.5% decrease respectively (rows 3 and 7). Rollouts of Legolas trained with the full sensing suite and only the IMU are visualized in Sec. E. The improved odometry predictions clearly show Legolas is capable of closely estimating the ground truth trajectory while only using the IMU leads to degraded behavior.

## 4.5 Real-World Lite3 Deployment

Legolas, when trained for the Lite3, is able to deploy onto real-world hardware as demonstrated in Fig. 4. As visualized in Fig. 4b, Legolas tracks the ground-truth trajectory of the robot, showcasing that Legolas is able to deploy across multiple robots. A visualization of the physical robot with ground truth and predicted trajectories in the scene is shown in Fig. 4a.

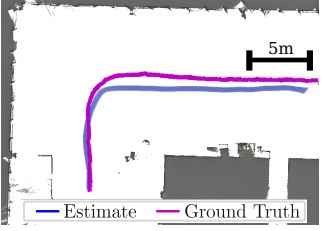

(a) Predicted and ground truth trajectory imposed on a map.

(b) Lite3 deployment with predicted and ground truth trajectories visualized.

Figure 4: **Legolas deploys across robotics platforms.** After collecting a new dataset of Lite3 rollouts in simulation and training a new model, Legolas successfully deploys to the new platform. The reconstructed trajectory is visualized from the top-down in Fig. 4a and in the real-world in Fig. 4b.

## 5 Conclusion

In this work, we introduced Legolas, a leg-inertial model capable of achieving odometry prediction at a level near visual odometry without explicit modeling of the robot or the environment. Legolas was trained by first collecting a dataset in a simulator to take advantage of its ability to parallelize and produce a diverse dataset. After this, Legolas is trained across multiple quadruped embodiments with successful real-world transfer for both. Through real-world experiments in both indoor and outdoor scenarios, we demonstrated Legolas's capabilities of estimating odometry at an accuracy higher than both filtering-based and other learning-based baselines.

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

## Appendix – Legolas: Deep Leg-Inertial Odometry

In this Appendix, we include additional details about the following:

## A    Limitations

We provide a comprehensive overview of potential failure modes and limitations, along with strategies for future improvements. (1) Legolas is locomotion-specific. Through testing, it has been found that Legolas's odometry prediction becomes poor when the locomotion policy used during deployment is different than the one used in training. A potential solution to this could be by collecting a diverse set of walking gaits in simulation and using this as a larger, more diverse source to train Legolas. (2) Legolas has no immediate solution for removing accumulating drift. Integrating the predictions of Legolas into a SLAM system as an odometry estimate, which is very common across modern SLAM systems [48, 42] could fix this. (3) Our proposed method is constrained by the fidelity and capabilities of the simulator. These limitations include simple foot-contact physics, limited sensing models resulting in poorer sim-to-real transfer, and a lack of dynamic obstacles in our current simulation framework.

## B    Further Training and Data Collection Details of Legolas

Training and data collection details are provided to improve the reproducibility of Legolas.

### B.1    Training Details

During deployment, it was found that the default 1D ResNet architecture [41] had too many parameters, causing slower than desired model prediction on real-world hardware. Fine-tuning of the model, by reducing the number of residual layers from 3 to 2 decreased the number of parameters in the network from 4.7M to 1.5M allowing for successful deployment. As shown in Tab. 3, this led to no substantial change in the relevant metrics.

| # | Model Parameter Sized | RPE@1m $\downarrow$ | ATE$_u$ $\downarrow$ |
|---|---|---|---|
| 1 | 4.7M | **0.045** | 0.017 |
| 2 | 1.5M (Ours) | 0.050 | **0.016** |

Table 3: **Effect of Model Size on Validation Performance.** While decreasing the size of the model reduced model performance in RPE@1m, ATE$_u$ improved slightly when using the smaller model. Due to the changes not substantially changing the relevant metrics, but boosting on-board speed, the smaller model was utilized for deployment.

### B.2    Data Collection

**Simulator Ground-Truth Data:** The ground truth sensor data is generated from the simulator Isaac-Gym with the real-time physics engine 'PhysX.' PhysX simulates the robot's configuration by integrating forces, velocities, and constraints based on the robot's properties (such as mass, and kinematic configuration) and environmental factors (such as friction, gravity, and other applied forces).

During each simulation step, PhysX updates the robot's state, including its position, orientation, and velocities, and accurately resolves any interactions with the environment. This detailed simulation data is then made available within Isaac-Gym, allowing seamless access to the ground-truth sensor data for use in various applications. Slippage is handled through PhysX through the coulomb friction model [49].

**Additive Noise:** We utilize the noise model from Rudin *et al*. [50] where the noise is sampled from a uniform distribution and then added to the observations. Tab. 4 includes the relevant minimum and maximum values of this distribution.

| Observation | Min and Max Values of Unif. Distribution |
|---|---|
| IMU Gyro | $\pm0.2$ rad/s |
| IMU Pitch and Roll | $\pm0.05$ rad |
| Command Velocity X/Y | $\pm0.0$ m/s |
| Command Velocity Angular | $\pm0.0$ rad/s |
| Joint Angle | $\pm0.01$ rad |
| Joint Velocity | $\pm1.5$ rad/s |
| Policy Action | $\pm0.0$ rad |

Table 4: **Sensor Noise.** Additive noise is sampled over a uniform distribution from the minimum and maximum values given. These ranges were found to be more noisy than their real-world counterparts.

**Simulator Dataset Collection:** The only difference in Legolas for training the pose prediction model $\mathcal{P}$ for the Go2 and Lite3 occurs during dataset collection, the 'urdf' file that describes the robot's kinematics is changed for the respective robots.

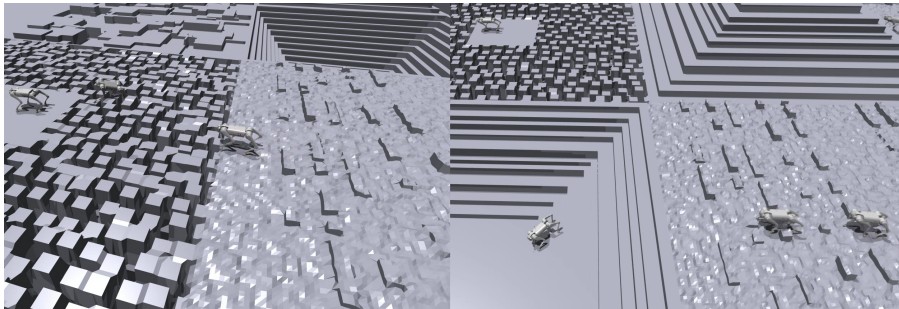

Figure 5: **Snapshot of robots collecting trajectories in simulation as performed in Sec. 3.1.**

To collect a given trajectory, a robot follows its policy $\hat{\pi}$ with a random velocity command with a target linear velocity uniformly sampled from $[-2.0, 2.0]$ $\frac{m}{s}$. A target linear velocity between $[-0.15, 0.15]$ $\frac{m}{s}$ is instead set to 0 in order to add standing and rotating in-place behaviors into our dataset. A target angular velocity in yaw is then uniformly sampled from $[-\pi, \pi]$ $\frac{rad}{s}$. If an angular velocity is chosen between $[-0.05, 0.05]$ $\frac{rad}{s}$, the angular velocity is set to 0 to create more walking forward behaviors in our dataset as this is a common motion in the real world. This large range of linear and angular velocities covers the full range of motions possible with $\hat{\pi}$. A visualization of the training environment can be found in Fig. 5.

## C  Training Loss Details

During training and deployment, different rotation representations were utilized. Using a 6D representation [39] during training improved performance on relevant metrics on the validation split as demonstrated in Tab. 5. Testing was performed against a yaw-pitch-roll representation and a combined translational representation. The combined translation representation is a logarithm map

| # | Rotation Representation | RPE@1m $\downarrow$ | ATE$_u$ $\downarrow$ |
|---|---|---|---|
| 1 | Yaw-Pitch-Roll | 0.233 | 0.023 |
| 2 | Log($SE(3)$) Loss | 0.085 | 0.022 |
| 3 | 6D [39] (Ours) | **0.050** | **0.016** |

Table 5: **6D rotation improves validation metrics.** Predicting changes in the 6D rotation produced better odometry prediction (RPE@1m decreased by 78.5%) when compared to predicting the changes in yaw, pitch, and roll.

of SE(3) with a velocity vector and axis-angle representation for rotation [51]. During training, $\Delta P_R$ is a 6D representation, and the ground-truth incremental motion of the robot is converted into a 6D representation. Otherwise, during deployment and data collection, a matrix representation is utilized.

# D  Robotics Experiment Details

The average trajectory length for the indoor and outdoor datasets are $24m$ and $325m$ respectively. The indoor dataset consists of scenes in an office building with flat, carpeted and stone floors as well as a stairs trajectory. The outdoor dataset consists of urban scenes on concrete, dirt, and grass.

Legolas is deployed on real-world hardware with a frequency of 50 Hz to match the frequency used in the simulator. However, with the optimizations provided in Sec. B.1, the model is capable of being run at up to 600 Hz on the Jetson Orin Nano.

During deployment of Legolas, the incremental motion of the robot is predicted at every time step. To estimate the trajectory of the robot, these predictions are accumulated. After converting the output of the models to an $SE(3)$ matrix at some arbitrary step i, $S_i$, the state at the current step, j, can be computed as $S = S_j S_{j-1}...S_0$.

## D.1  Visual-Inertial Odometry Deployment Details

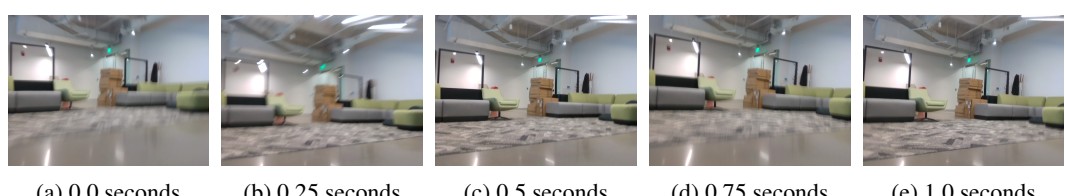

| (a) 0.0 seconds | (b) 0.25 seconds | (c) 0.5 seconds | (d) 0.75 seconds | (e) 1.0 seconds |

Figure 6: **Quick motions and large changes in rotation occuring over one second.** These quick motions cause degraded image captures on the quadruped affecting the deployment of the visual baseline. Changes in pitch (Fig. 6a,6b,6c,6d,6e) and roll (Fig. 6a and Fig. 6e) are notable.

VINS-Fusion found successful deployment when carefully tuned using tools including 'imu_utils' [43, 52] and 'Kalibr' [44]. imu_tools is utilized to retrieve relevant IMU statistics used by both the EKF and VINS-Fusion baselines such as the gyroscope's and accelerometer's bias and noise. Kalibr was used to find the extrinsic matrix of the camera frame with respect to the IMU frame and to find the timing offset between the camera and the IMU. With these steps, more manual fine-tuning of the parameters was required for deployment.

However, direct deployment to outdoor environments of the previously tuned and measured parameters failed. We found that quick motions and rotations such as the one visualized in Fig. 6 became exaggerated on a quadruped robot. Furthermore, interactions between the sun and the onboard camera caused degraded depth prediction and tracking. Using a polarized film with a metal shield around the camera reduced this error mode, but still didn't lead to consistent results as shown in Fig. 7.

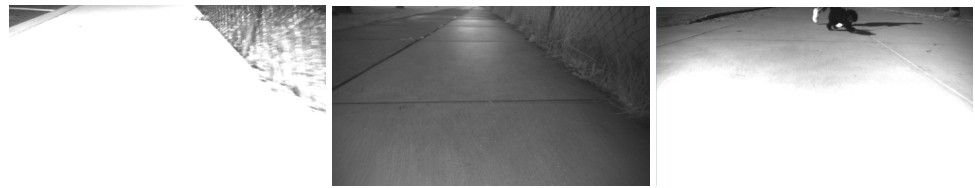

(a) Stereo camera without shield and full cloud coverage.    (b) Stereo camera with shield and full cloud coverage.    (c) Stereo camera with shield and no cloud coverage.

Figure 7: **A camera shield is necessary for VINS-Fusion deployment.** Without the camera shield, the stereo camera equipped on the robot fails to capture the scene due to interference. However, we still find issues with outdoor deployment of VINS-Fusion if the environment is sunny even when equipped with shielding.

### D.2 Improving VIO with Legolas

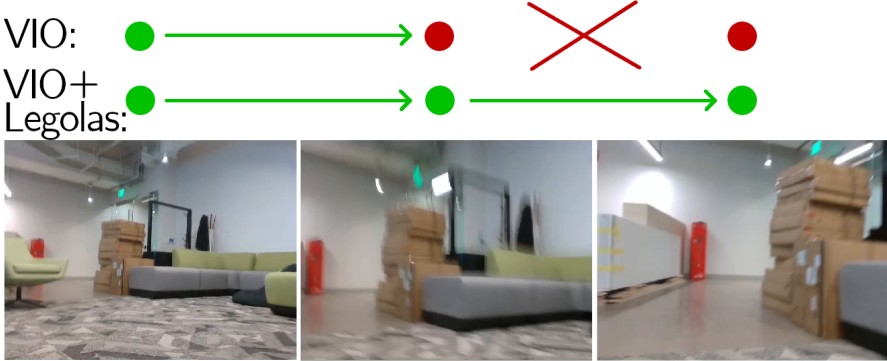

Figure 8: **Legolas improves VIO:** RTAB-Map as a visual baseline loses tracking while the robot is moving. However, with Legolas integration, RTAB-Map is able to continue mapping.

As Legolas predicts both an odometry estimate as well as a covariance, these estimates can be used in downstream SLAM applications. RTAB-Map [48] is utilized in this experiment as it allows for easy integration of odometry estimates from outside sources. In Fig. 8, adding Legolas to RTAB-Map allows for the method to continue mapping even when visual tracking fails. The complementary video for this experiment is available at: https://learned-odom.github.io/index.html#rtabmap.

### D.3 Details on Training Test Dataset

A visualization of the collected indoors dataset is visualized in Fig. 9. Ground truth was collected with a motion capture and a LiDAR-SLAM system.

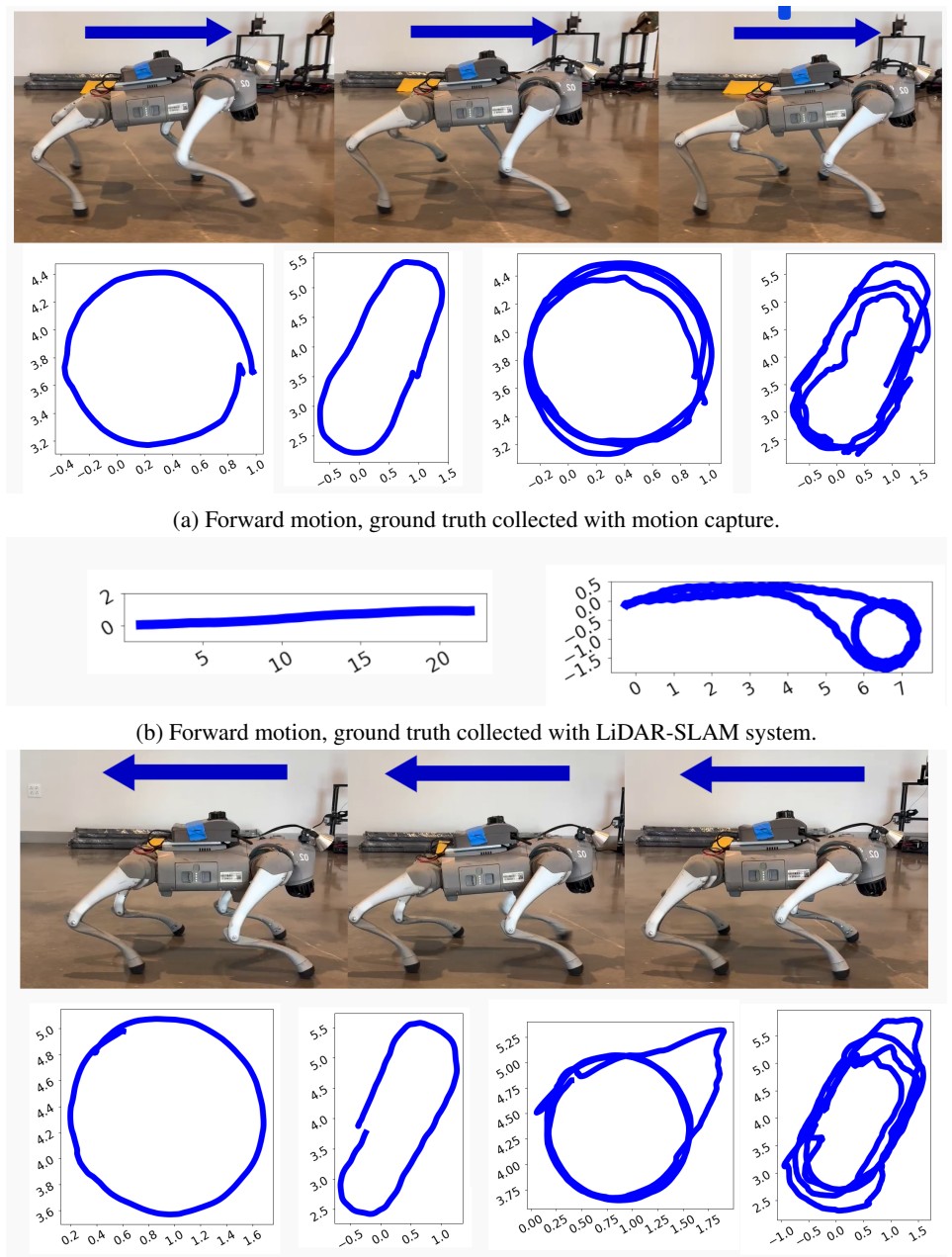

(a) Forward motion, ground truth collected with motion capture.

(b) Forward motion, ground truth collected with LiDAR-SLAM system.

(c) Backwards motion, ground truth collected with motion capture.

Figure 9: **Collected Indoor Test Dataset:** Units in meters. The indoor test dataset consists of 10 collected trajectories. 6 trajectories were collected with the Go2 moving forward. Of the collected forward trajectories, 4 obtained their ground truth from a motion capture system and 2 from a LiDAR-SLAM system. Four trajectories were collected with the Go2 moving backward with ground truth obtained from a motion capture system.

# E  Additional Trajectory Rollouts

**Additional validation rollouts:** Fig. 11 demonstrates the superior trajectory reconstruction of using the full sensing suite rather than just the IMU. The use of the full sensing suite allows Legolas to directly predict the incremental motions of the robot rather than rely on analytical models of the robot.

## E.1  Real-World Variance Predictions

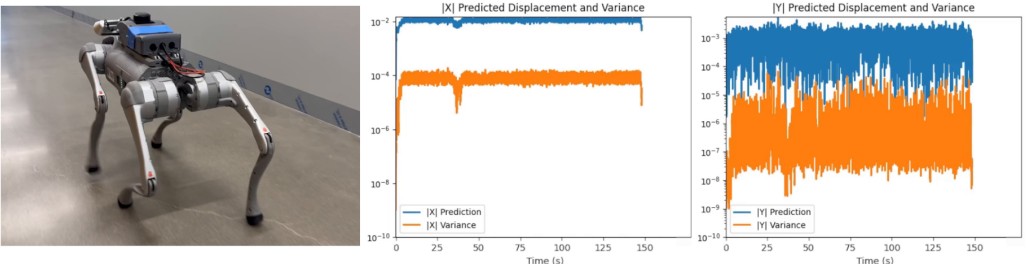

(a) **Building Scene:** The robot walks through hallways and carpeted office areas. The predicted magnitude of the displacements and variances are shown. Notably the Y variance and displacements are much smaller than the forward-facing X direction as the displacements from swaying are much smaller.

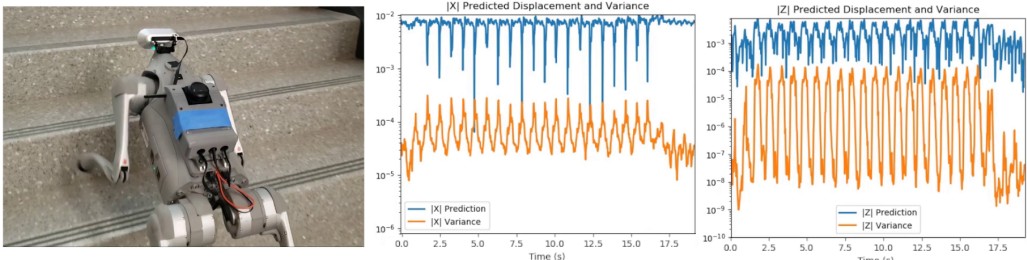

(b) **Stairs Scene:** the robot goes up stairs. The predicted magnitude in the forward-facing X and upwards Z direction are cyclical with a matching variance.

Figure 10: **Predicted variance and displacements.** Across a building scene Fig. 10a and going up stairs Fig. 10b, results for the predicted magnitude in displacements and variances are shown.

A qualitative analysis of the robot's predicted variance across two scenes is examined. A companion video showcasing these rollouts, as well as the confidence intervals, can be found at https://learned-odom.github.io/index.html#variance.

**Building Scene Fig. 10a:** In this scenario, the Y displacement and variance are much smaller than the X displacement. This matches what is expected as the robot is moving forward, and this would be dominated by an X displacement with a smaller Y displacement from swaying. After completing the first hallway, the robot turns sharply and causes a dip in displacements, which are matched with a dip in variance as the robot is moving slower. The lower variances when the robot is moving slower can also be observed at the start of the plot as the robot starts to move.

**Stairs Scene Fig. 10b:** While going up the stairs the X and Z displacements are very cyclical with their peaks being matched with peaks in variance occurring wherever the displacement changes quickly. When the robot reaches the new floor, variances for X and Z decrease as there is the robot's locomotion is more stable.

**Mean relative variance increases with task difficulty.** As visualized in Fig. 10, the variance magnitude changes with the robot's speed; slower movement narrows the range of reasonable predictions. Differences in error should also occur with varied robot locomotions. Tab. 6 utilizes mean variance magnitudes divided by prediction magnitudes across scenarios. This metric represents the certainty of the network in its prediction. During stable locomotion (Building and the last 2 seconds of Stairs), this ratio is lower meaning that the network is more certain of its prediction. During the

| Environment | $\frac{\lVert\mathbf{X}\rVert\text{ Variance}}{\lVert\mathbf{X}\rVert\text{ Prediction}}$ | $\frac{\lVert\mathbf{Y}\rVert\text{ Variance}}{\lVert\mathbf{Y}\rVert\text{ Prediction}}$ | $\frac{\lVert\mathbf{Z}\rVert\text{ Variance}}{\lVert\mathbf{Z}\rVert\text{ Prediction}}$ |
|---|---|---|---|
| **Building** | 0.0054 | 0.0032 | 0.0011 |
| **Stairs (Full)** | 0.0213 | 0.0184 | 0.0061 |
| **Stairs (Last 2 Seconds)** | 0.0039 | 0.0029 | 0.0008 |

Table 6: **Mean relative variance increases with the difficulty of the task.** The mean ratio of the variance magnitude over the prediction magnitude is lower for simple walking tasks (Building and last 2 seconds of Stairs). This ratio increases when the robot is climbing up stairs in the full Stairs environment.

full stairs rollout, where quick motions occur, the ratio increases as predicting changes in motion on stairs is more challenging.

**Additional real-world rollouts:** Supplementing the rollout given in Fig. 3a, additional rollouts are visualized. Fig. 12 demonstrates the robot moving in a straight line in an indoor scene. In this scenario Legolas tracks the straight line while VINS-Fusion suffers from losing visual tracking due to reflections on the floor. Fig. 13 demonstrates the robot moving in a double circular pattern in an indoor environment. In this scenario, Legolas is able to track the ground-truth. The EKF baseline is capable of reproducing a similar shape up until the 1st completed loop where its trajectory reconstruction degrades.

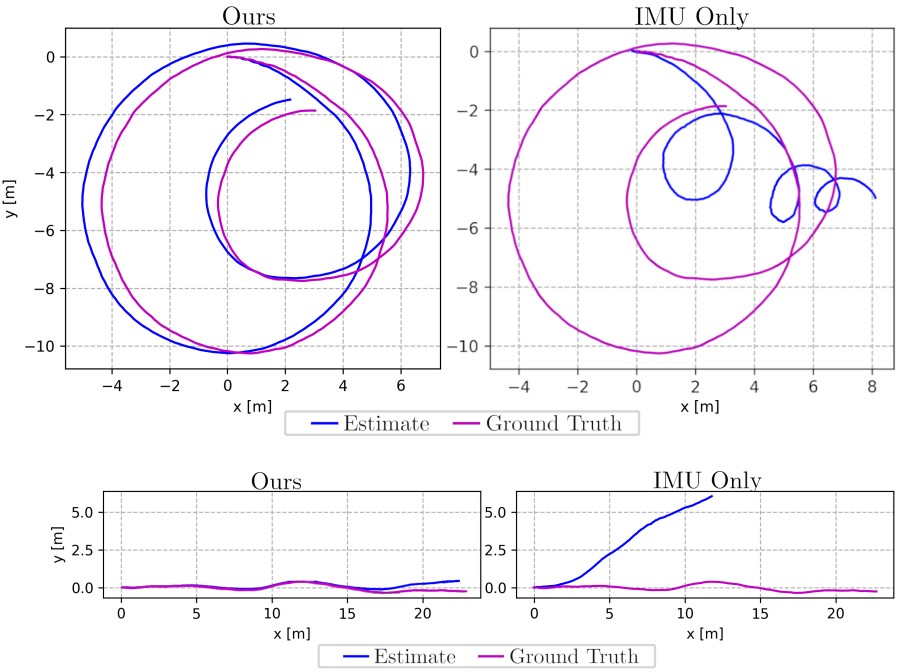

Figure 11: **Top-Down visualization of rollouts of Legolas trained with different sensing modalities.** Legolas demonstrates improved odometry prediction through its use of the full sensing suite on the robot. Previous work has attempted to estimate displacements through learning with only the IMU sensor, this produces inferior predictions.

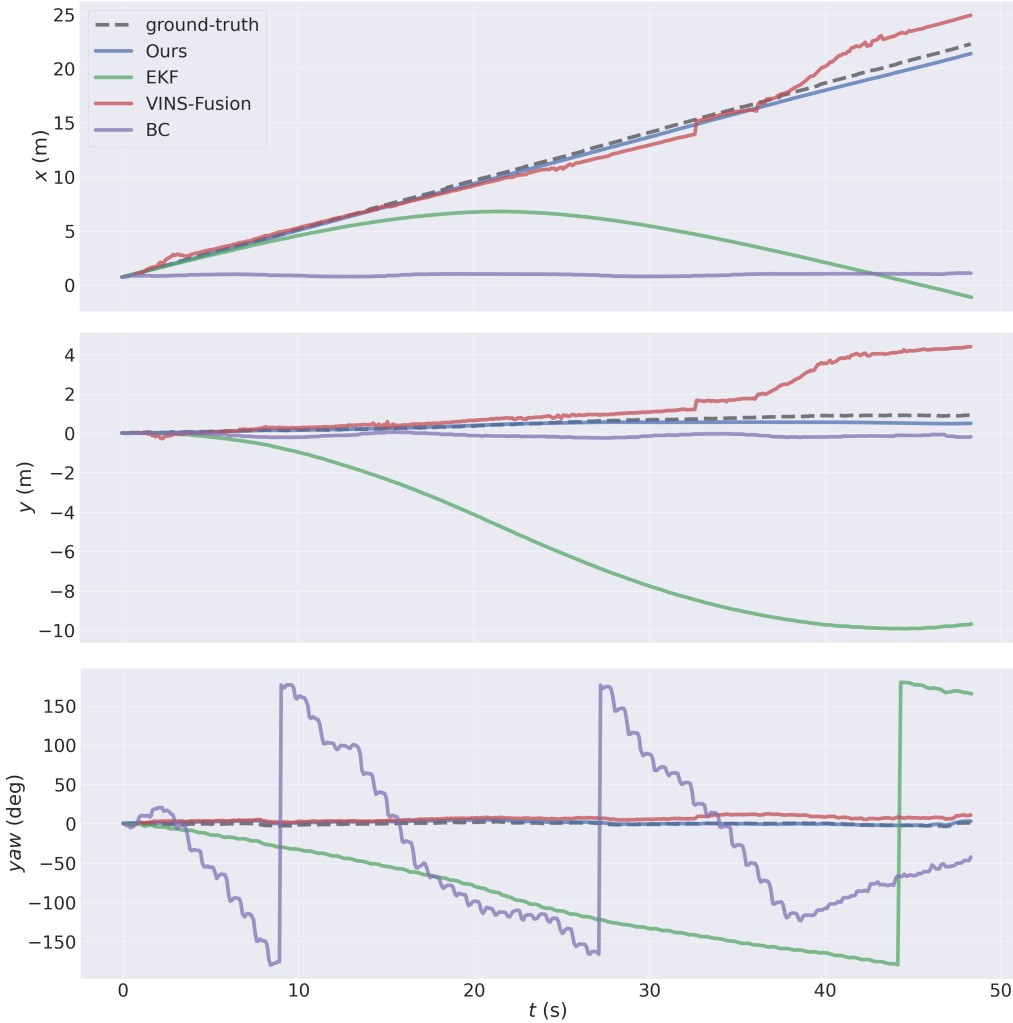

Figure 12: **Straight line rollout.** In this scenario the robot moves in a straight line for approximately 23 meters. Legolas tracks the trajectory, while VINS-Fusion suffers from losing visual tracking around $t(s) = 37$ and the estimated position of the robot jumps.

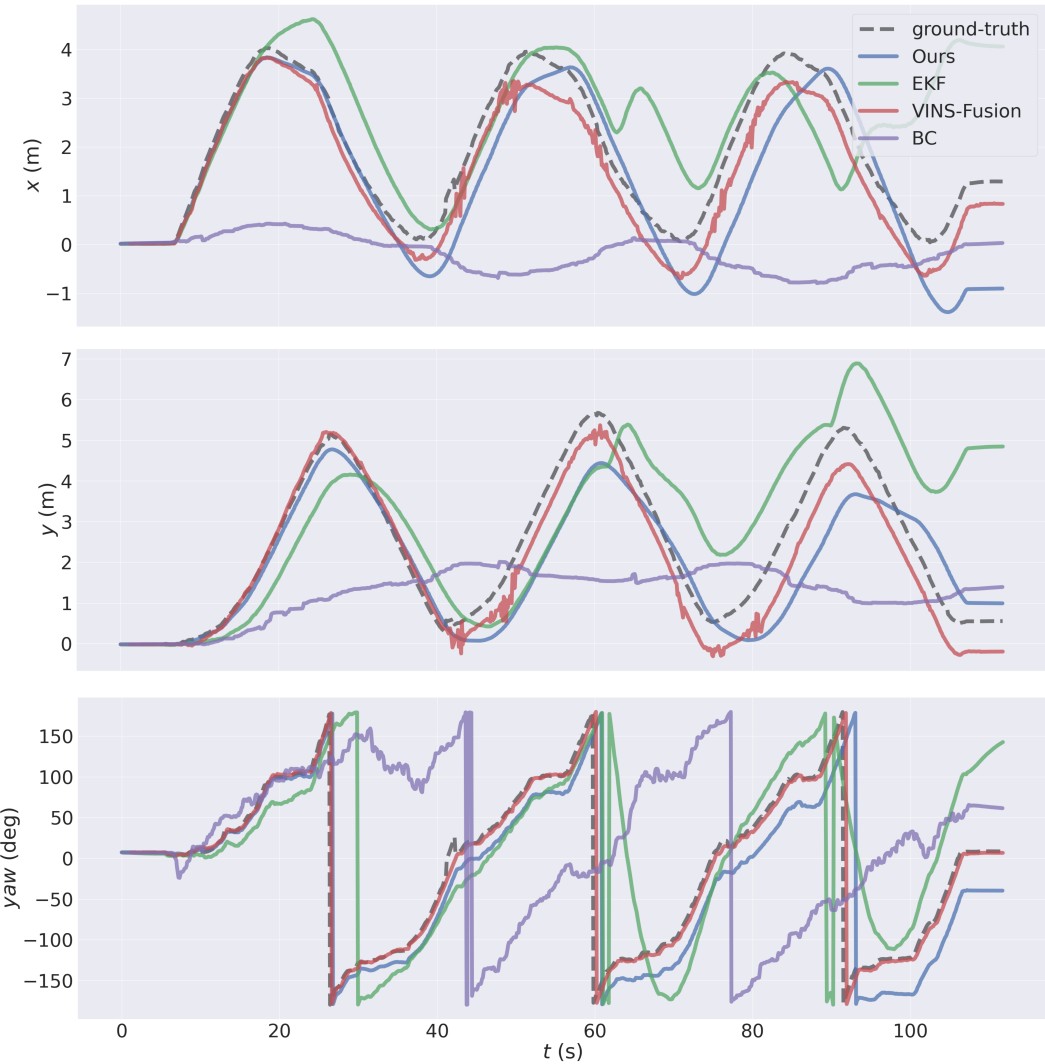

Figure 13: **Circular rollout**. In this scenario the robot moves in a circle two times, both VINS-Fusion and Legolas are capable of tracking the ground-truth trajectory. EKF tracks closely until the first loop is completed. BC is able to track the heading, but poorly track x and y positions.

