# OpenReview forum: "Legolas: Deep Leg-Inertial Odometry"
_robot-learning.org/CoRL/2024/Conference — CoRL 2024_

### Official Review · Reviewer_8eXK · 2024-07-06
**Interesting work but many questions**

**Originality:** 3
**Technical Quality:** 3
**Clarity Of Presentation:** 3
**Potential Impact:** 3
**Recommendation:** 3
**Confidence:** 5

**Review:**

I think this paper is well written and proposes an interesting method. I have a few questions and concerns which I feel the auothors should address.

1. The paper needs to explicitly describe the inputs and outputs of the neural network and their dimensionality in the body of the paper. I realize there is some discussion in the supplementary material about the output representation, but this needs to be in the main body of the paper. What are the inputs to the network? The authors have written proprioceptive signals but does this mean joint positions/velocities/torques? Is it for all legs? What are the actions?

2. The results that the authors obtain for the Bloesch et al method which they label as EKF, are much worse than anything I have experienced with this method. It makes the paper much less convincing when the baseline is significantly worse than previously reported for example in Buchanan et al. "Learning inertial odometry for dynamic legged robot state estimation"

3. I appreciate the authors have tested with different locomotion policies and even different robots. They have also discussed how this shows promise but does not immediately mean the method is invariant to locomotion. This is good. I just have a question about why Ours w/ Go1 fails in the outdoor scene? Does the robot slip significantly?

4. How would the authors consider simulating slip? In Buchanan et al. they used real data and created real slip scenarios. As I understand it, this is quite difficult to do in simulation.

**Quality Of The Limitations Section:**

3

**Questions For Rebuttal:**

1. The inputs and outputs of the neural network must be much more clearly explained.

2. The poor performance of the EKF method should be more clearly explained

**Robotics Focus:**

4

**Summary Of Paper:**

This paper introduces a sim-to-real odometry prediction system for legged robots using IMU+leg odometry+velocity commands.

**Summary Of Recommendation:**

I lean towards accept and ask that the authors provide the additional explanations that I have requested.

---

### Official Review · Reviewer_KiPa · 2024-07-14
**Quality of Experimental Results Insufficient**

**Originality:** 2
**Technical Quality:** 2
**Clarity Of Presentation:** 2
**Potential Impact:** 2
**Recommendation:** 2
**Confidence:** 5

**Review:**

**STRENGTHS**

-	The motivation is very well laid out. Even tough exact kinematic models may be possible to produce, the arguments presented why it is easier to adopt a data-driven approach seem valid to me.
-	The authors perform real-world experiments with different robot models. That increases confidence that the proposed method works on different models. Unfortunately, the quality of the method is only evaluated for one of the models.

**WEAKNESSES**

-	Methodologically, this paper does not add much compared to the discussed prior work of Cioffi et al. The only difference is to train on simulated data instead of real-world data captures.
-	The results are very weak. At the current stage, the outdoor results need to be discarded because the ground-truth does not have sufficient accuracy. The indoor results are worse than VINS-Fusion, and show inconsistency between the priors that shoud be learned and those demonstrated (training with Go1 is better than with the correct kinematic model).
-	One of the motivations for the work is that visual baselines would be prone to failure. This failure should be better proven. So far the authors simply say “the visual baseline fails to deploy outdoors.”. As somebody who already used visual-inertial-SLAM outdoors on legged robots, that seems more like a user-error. Most outdoor MAVs also use visual-inertial odometry. The authors need to better demonstrate the failure cases. Especially since this work is about odometry, loosing tracking should not be counted as overall failure but just temporarily bridged with a constant velocity model until tracking is established again. This will still create error in the evaluation, but is more informative than no number at all.
-	The Lite3 Deployment is fine as a demonstration, but the authors should abstain from words such as “successful” or “accurate” without quantative evaluation over multiple longer trajectories.
-	What is the real application for IMU-legged odometry that discards vision or LiDAR as input? I think the authors should be more transparent that the most obvious application for their method is military, where a robot should operate in the dark and without active sensors.



**DETAILED COMMENTS**

-	Line 25-26: This introduction of odometry is wrong. Odometry is not used to reduce drift. Odometry is what is causing drift. Loop-closures or external tracking systems are used to compensate for drift from odometry (which can be visual, kinematic, IMU, LiDAR, or based on something else).
-	Line 29: what about “autonomous car navigation” is “far beyond robotics”?
-	Line 61: what do the authors mean with manual fine-tuning? editing neural network parameters by hand? Who does something like that?
-	Line 61: The current results do not allow to make the claim that this method generalizes across platforms, because quantitative evaluation is only done with 1 robot.
-	Line 90: There is still no guarantee that this method does not go out-of-distribution. It is just less likely.
-	It would help to have the evaluation scenarios and trajectories visualized in a figure. For example, it is unclear how diverse the environments are, how long the trajectories, and what kind of terrains there are present. For example, do the robots walk up and down stairs, one of the main use cases of legged robots?
-	Table 1: The differences between methods are mostly below 30cm. This raises the question how accurate the generated ground truth is. For outdoors, the paper describes the use of GPS for ground truth. GPS only provides around 1m accuracy. This would mean that the results in the outdoor column are meaningless, and the comparison up to mm scale is really a misrepresentation of the measurement accuracy.
For Indoors, the paper is not clear whether the ground-truth is always generated from motion capture, or if large parts of the ground-truth are obtained through LiDAR-SLAM. LiDAR also only provides a few cm accuracy, so this groundtruth should as well not be used to compare measurements up to a mm level. Here, however, the results do not really change if numbers are rounded to cm level.
-	Since the algorithm predicts variance, an evaluation of how accurate this variance is would be interesting.

**Quality Of The Limitations Section:**

3

**Questions For Rebuttal:**

see review above.

**Robotics Focus:**

4

**Summary Of Paper:**

This paper continues prior work to learn odometry estimates from IMU and leg encoders for quadruped robots. The innovation on top of prior work is to train on simulated data.

**Summary Of Recommendation:**

The rebuttal significantly improved the accuracy of the experimental results. The mix of the experiments is now a bit chaotic with different camera placements, environments, etc. I think there is now sufficient evidence that this works in general.  I am however still not able to conclusively answer the questions "How useful is this compared to VIO with a correctly placed camera?" and "If, as the authors suggest, this needs to be combined with a visual SLAM system, does it really yield improvements over other odometry methods?"; I therefore raise my score to weak reject.

---

### Official Review · Reviewer_hmFN · 2024-07-15
**A good demonstration of the progress of simulation-based learning solutions to classical problems**

**Originality:** 4
**Technical Quality:** 3
**Clarity Of Presentation:** 4
**Potential Impact:** 3
**Recommendation:** 3
**Confidence:** 5

**Review:**

**Summary**
I liked the paper as it presents a simple approach that capitalizes on the progress of learning-based methods and cheap simulation data, on a problem that is traditionally solved in an analytical, model-based way. The evaluation is sensible, though there are points to clarify further.

**Strengths**
The main strength is that the approach is simple and straightforward. This makes it easily adaptable to other robotic platforms. It also makes a lot of sense to focus on proprioceptive estimation, as this data is easier to simulate than other exteroceptive sensing (e.g. point clouds or, even worse, images).
The paper is well-written and easy to follow, with a good literature review of the related works.

**Weaknesses**
I don’t see significant weaknesses but I did find aspects that must be clarified further. Particularly some definitions, and connections to the standard legged odometry literature. I also found some conceptual problems in the evaluation, which I discuss in the questions for the rebuttal section.

**Quality Of The Limitations Section:**

2

**Questions For Rebuttal:**

- Please be more explicit about the loss definition for the incremental pose change. Currently it is not clear how the actual Euclidean error is obtained if the pose difference is defined on SE(3). I had to check the appendix to understand that the translation and rotation are decoupled and a special parametrization for rotations was used.
- Further, I wonder if using a different error definition (i.e metric) that couples rotation and translation (e.g. the logarithm map of SE(3)) could change the results, given that’s the standard procedure to define factors in SLAM or odometry estimation with factor graphs.
- Given that the method predicts the covariance of the estimate, it would be interesting to see the confidence bounds of the predictions, as this information can be used in downstream methods (e.g. SLAM). This could provide further insights on the consistency of the estimate.
- The “masking head” term introduced is doing nothing else than “zero velocity updates” in standard legged state estimation literature. I recommend making this connection because otherwise it’s an unnecessary rebranding of a widely studied concept. The $m$ variable introduced is also similar to the covariance increases during flying phases of the legs used in other papers, such as Bloesch (2012), which acts like a soft switch.
- Please explain how the IMU and other data is simulated. Which analytical models and noises did you use? How far are they from the nominal specifications of the real IMU and sensors used?
- Evaluation: Using the Umeyama method for odometry evaluation is not right and it is expected to produce lower errors. Umeyama makes sense when you evaluate trajectories estimated by a SLAM system, which introduces loop closures that change the reference frame of the trajectories. In this setting of odometry estimation, aligning against the initial pose is the only case that makes sense.
- Similarly, the ATE is not really a good metric to evaluate odometry, as it is expected to drift, and the numbers will get worse over time. The RPE is a better metric, because it effectively characterizes error per distance traveled, which encodes the amount of drift expected for the method.
- The real world deployment example on Lite3 is not really useful. Fig 4 doesn’t show a scale, so it’s difficult to understand how much the estimate drifted. Running a standard odometry system on a different robot platform is also expected to work. The question always is how much is the actual drift.
- Lastly, I wonder how much effort was made to make the VIO methods work. None of the trajectories look particularly challenging (and let’s note that not much information about the sequences was shown in the paper itself), so I’m a bit hesitant. Maybe testing against OpenVINS or explaining the failure cases in a better way could be a useful insight for the readers.

**Robotics Focus:**

4

**Summary Of Paper:**

The paper presents a learning-based method to estimate legged odometry using simulated IMU and proprioceptive data. Using different rollouts of a policy in simulation, a network is trained to predict incremental pose estimates, which are concatenated to produce a SE(3) pose. Results show significant improvements to other baselines that rely on exteroceptive data as well.

**Summary Of Recommendation:**

I think the paper is good and interesting, so I’m happy to accept it. But I strongly encourage the authors to address the previous comments, because these are weak aspects that people working in state estimation will very likely criticize otherwise, since they omit or rebrand some known ideas in the field.

---

### Author Rebuttal · Authors · 2024-08-12

Thank you everyone so much for your efforts to help improve our work! We have attached the updated rebuttal here.

---

### Decision · Program_Chairs · 2024-09-04

**Decision:**

Accept

**Comment:**

This paper received mixed initial reviews with the following summary of strengths and weaknesses.

Strengths:
- The approach is simple and straightforward which could make it easily adaptable to other robot platforms.
- The paper is well written and easy to follow.

Weaknesses:
- Novelty is limited, as a similar approach has been proposed before trained on real-world data.
- Further clarifications of the method are needed.
- The method should be better related to the previous literature in legged robot odometry estimation.
- The experimental results are weak and evaluation needs to be improved.
- The results for the VIO baseline in the experiments are questionable.

The author response and discussion has partially addressed the concerns of the reviewers. After some debate, the reviewers and AC concluded that the reasons for accept outweigh reasons for rejecting the paper and it can be recommended for acceptance.

However, the additional covariance evaluation did not address the questions/concerns by the reviewers appropriately (it seems there is a misunderstanding) and does not provide sufficient evidence if the variance follows the prediction error. This part must either be fixed entirely following the suggestions by the reviewers or removed from the paper. If it is fixed and included, it is suggested to move this experiment to the supplementary material to also keep within the page limit.

The revised paper violates the page limit by over 1 page. Please write some parts more compactly and move some parts (e.g. the covariance evaluation) to the supplementary material. If the paper gets accepted, the final paper must not violate the page limit.